# Non-rapid eye movement sleep determines resilience to social stress

**Brittany J Bush[1], Caroline Donnay[1], Eva-Jeneé A Andrews[1], Darielle Lewis-Sanders[1], Cloe L Gray[1], Zhimei Qiao[1], Allison J Brager[2], Hadiya Johnson[1], Hamadi CS Brewer[1], Sahil Sood[1], Talib Saafir[1], Morris Benveniste[1], Ketema N Paul[3], J Christopher Ehlen[1]\***

[1]Neuroscience Institute, Morehouse School of Medicine, Atlanta, United States; [2]Behavioral Biology Branch, Center for Military Psychiatry and Neuroscience, Walter Reed Army Institute of Research, Silver Spring, United States; [3]Department of Integrative Biology and Physiology, University of California, Los Angeles, Los Angeles, United States

**Abstract** Resilience, the ability to overcome stressful conditions, is found in most mammals and varies significantly among individuals. A lack of resilience can lead to the development of neuropsychiatric and sleep disorders, often within the same individual. Despite extensive research into the brain mechanisms causing maladaptive behavioral-responses to stress, it is not clear why some individuals exhibit resilience. To examine if sleep has a determinative role in maladaptive behavioral-response to social stress, we investigated individual variations in resilience using a social-defeat model for male mice. Our results reveal a direct, causal relationship between sleep amount and resilience—demonstrating that sleep increases after social-defeat stress only occur in resilient mice. Further, we found that within the prefrontal cortex, a regulator of maladaptive responses to stress, pre-existing differences in sleep regulation predict resilience. Overall, these results demonstrate that increased NREM sleep, mediated cortically, is an active response to social-defeat stress that plays a determinative role in promoting resilience. They also show that differences in resilience are strongly correlated with inter-individual variability in sleep regulation.

**\*For correspondence:** jehlen@msm.edu

**Competing interest:** The authors declare that no competing interests exist.

## Editor's evaluation

This well-written, convincing report provides new insights for neuroscientists studying sleep architecture and stress sensitivity. A particularly important conclusion is that differences in sleep architecture before chronic social defeat stress may serve as a predictive biomarker of stress resilience.

## Introduction

The links between sleep, neuropsychiatric illness, and responses to stress have been extensively documented, but are poorly understood. Sleep disorders are debilitating features of neuropsychiatric conditions (for a review see *Wulff et al., 2010*), and social stress is known to cause or exacerbate both sleep and neuropsychiatric disorders (*Meerlo et al., 1997*; *Meerlo et al., 2001*; *Germain et al., 2003*; *Huhman, 2006*). Notably, both self-reported sleep disturbances and objective polysomnographic recordings are predictive of the development of stress-induced disorders such as posttraumatic stress disorder, depression, and anxiety (*Mellman et al., 2002*; *Koren et al., 2002*; *Germain et al., 2006*; *Bryant et al., 2010*; *Sheaves et al., 2016*; *Ben Simon and Walker, 2018*). In some cases, both humans and animals show resilience to the adverse effects of stress. The degree of resilience is highly variable between individuals for reasons that are not entirely clear (*Galatzer-Levy et al., 2018*;

**eLife digest** To many of us, it may seem obvious that sleep is restorative: we feel better after a good night's rest. However, exactly how sleep benefits the brain and body remains poorly understood. One clue may lie in neuropsychiatric disorders: these conditions – such as depression and anxiety – are often accompanied by disrupted sleep. Additionally, these neuropsychiatric disorders are frequently caused or worsened by stress, which can also interfere with sleep. This close association between stress and sleep has led some to hypothesize that sleep serves to overcome the adverse effects of stress on the brain, but this hypothesis remains largely untested.

One type of stress that is common to all mammals is social stress, defined as stress caused by social interactions. This means that mice and other rodents can be subjected to social stress in the laboratory to test hypotheses about the effects of stress on the brain. Importantly, in both animals and humans, there are individual differences in resilience, or the ability to overcome the adverse effects of stress.

Based on this information, Bush et al. set out to establish whether sleep can regulate resilience to social stress in mice. When the mice were gently kept awake during their normal sleep time, resilience decreased and so the mice were less able to overcome the negative effects of stress. Conversely, increasing sleep, by activating an area of the brain responsible for initiating sleep, increased the mice's resilience to social stress. Thus, Bush et al. showed that changes in sleep do lead to changes in resilience.

To find out whether resilience can be predicted by changes in sleeping patterns, Bush et al. studied how both resilient mice and those susceptible to stress slept before and after social stress. Resilient mice would often sleep more after social stress; meanwhile, few changes were observed in susceptible mice. Surprisingly, sleep quality also predicted resilience, with resilient mice sleeping better than susceptible mice even before exposure to social stress. To determine whether the differences in sleep that predict resilience can be detected as brain activity, Bush et al. placed electrodes in two regions of the prefrontal cortex – a part of the brain important for decision-making and social behaviors – to measure how mice recovered lost sleep. This experiment revealed that the changes in sleep that predict resilience are prominent in the prefrontal cortex.

Overall, Bush et al. reveal that sleeping more and sleeping better promote resilience to social stress. Furthermore, the results suggests that lack of sleep may lead to increased risk of stress-related psychiatric conditions. Humans are one of the few species that choose to deprive themselves of sleep: Bush, et al. provide evidence that this choice may have significant consequences on mental health. Furthermore, this work creates a new model that lays the groundwork for future studies investigating how sleep can overcome stress on the brain.

*Potegal et al., 1993*). The experiments reported here test two hypotheses: (1) that sleep amount plays a determinative role in the response of an individual to stress and (2) that inter-individual variability in sleep response to stress result in individual differences in resilience.

To investigate these hypotheses, we used a well-established model of social conflict in mice wherein rodents are socially stressed in the home cage of a larger conspecific. This model is frequently used to model core aspects of human pathologies with high face, construct, and etiological validity (*Huhman et al., 1990*; *Berton et al., 2006*; *Vialou et al., 2010*; *Krishnan et al., 2007*; *Hammamieh et al., 2012*). An important aspect of this social-defeat model is that not all socially stressed animals respond equally well—some mice are susceptible, and others are resilient to the effects of stress (*Krishnan et al., 2007*). These differences in the behavioral responses to social-defeat stress provide the opportunity to compare brains of susceptible and resilient animals and investigate the brain mechanisms contributing to these behavioral phenotypes. Within this model, multiple functionally interconnected brain regions have a demonstrated role in determining resilience. These regions include the nucleus accumbens (*Berton et al., 2006*; *Vialou et al., 2010*), amygdala (*Jasnow and Huhman, 2001*), hippocampus (*Wagner et al., 2013*), and medial prefrontal cortex (mPFC; *Hultman et al., 2016*). To determine the role of sleep in resilience to social-defeat stress, we have experimentally altered sleep amount; to examine the role of inter-individual variability in sleep, we have conducted detailed examinations of sleep and sleep-response to social-defeat stress in both resilient and susceptible mice.

## Results

We first looked to determine if sleep was necessary for resilience to social-defeat stress. One cohort of mice received daily bouts of social stress at the onset of darkness. In these mice, social interaction (measured by interaction ratio, see Materials and Methods for details) was assessed 1 day before and 1 day after social-defeat stress (*Figure 1A*). As expected (*Krishnan et al., 2007*), half of these mice were resilient (interaction ratio >1.1) to the effects of social-defeat stress (4 of 8 mice were resilient; *Figure 1B and C*; two susceptible—interaction ratio <1, two undefined—interaction ratio between 1.1 and 0.9). A second cohort of mice was sleep restricted (8 h; n.b. sleep restriction is synonymous with sleep deprivation. Animals unavoidably obtain small amounts of sleep during these sleep-limiting interventions; therefore, the term restriction may be more accurate) at light onset during the ten consecutive-days of social-defeat stress (*Figure 1A*). In contrast to the mice without sleep restriction, no sleep-restricted mice were resilient; furthermore, all sleep-restricted mice showed decreased social interaction after social-defeat stress (*Figure 1B and C*; five susceptible, two undefined – interaction ratio between 1.1 and 0.9). Differences in social interaction were not due to changes in ambulatory activity, as all treatment-groups showed similar levels of activity during the social-interaction testing (*Figure 1E*). The number of aggressive encounters across the social-defeat stress paradigm are shown in *Figure 1—figure supplement 1*. In a separate cohort of sleep-deprived mice we assessed fecal corticosterone to estimate stress levels (*Figure 1D*). These findings show that hypothalamic-pituitary adrenal axis stress-pathways are not significantly activated by sleep restriction (*Figure 1B*); However, this method assessed corticosterone over a 24 – hour period. It is possible that acute responses measured over a shorter period, and closer to the exact time of stress exposure, may differ. Collectively, these results demonstrate that sleep restriction increases susceptibility to social-defeat stress and provides evidence that sleep is necessary for resilience to social-defeat stress.

To further investigate the role of sleep in resilience, we increased sleep amount using a novel method that avoids the use of somnogenic drugs (*Figure 2A*). Because sleep is initiated by GABAergic cells projecting from the preoptic area (POA), we used a designer-receptor exclusively activated by designer-drugs (DREADD) to activate these POA neurons. Four weeks after delivering DREADD to the POA by adeno-associated virus, clozapine-n-oxide (CNO) was used to activate the DREADD and increase sleep. This was validated in a separate cohort of mice at zeitgeber time 10 (ZT, ZT 12=lights off, *Figure 2C*), where a single injection of CNO increased non-rapid eye movement (NREM) sleep for approximately 6 hr. Single, daily injections (administered ZT 1–2) of CNO across the 10-day social-defeat paradigm enhanced total sleep by 79.3±12.5 (mean ± SEM) min per day over control mice, (*Figure 2B*) including 114.6±14.2 min per day of NREM sleep. Part of this NREM increase occurred at the expense of rapid eye movement (REM) sleep; REM sleep was reduced by 31.6±4.7 min per day. Mice expressing the active DREADD (increased sleep) had significantly increased interaction ratios (increased resilience) after social-defeat stress, when compared to CNO-treated control mice expressing EGFP (*Figure 2E*). Notably, all mice expressing active-DREADD (increased sleep) were resilient (interaction ratio >1.1), whereas only half of DREADD-control mice were resilient. These results provide evidence that increased sleep confers resilience and has a protective influence during exposure to social-defeat stress; supportin our hypothesis that sleep plays a determinative role in resilience to social-defeat stress.

We next looked for the differences in sleep amount and architecture between resilient and susceptible populations. Analysis of sleep before and after social-defeat stress revealed dramatic changes in post-defeat sleep, but only in resilient mice (*Figure 3A, B and C*). Both the REM and NREM sleep of resilient mice were increased throughout the active/dark period following defeat, when compared to baseline sleep (*Figure 3C*). These sleep changes after defeat included an increase in total sleep amount over the 24 hr period (*Figure 3D*) and a reorganization of sleep from typical baseline patterns. Indeed, most of the post-defeat changes in total-sleep occurred during the active/dark period (*Figure 2D*). Control mice exposed to novel cages over 10 days did not have this reorganization of REM and NREM sleep. Instead, control mice had a modest increase in NREM and REM sleep during the dark period (*Figure 3—figure supplement 1*), showing that the effect of a novel environment on sleep was minor. NREM slow-wave activity (power 0.5–4 Hz) and theta activity (6–10 Hz), were also altered in resilient mice (*Figure 3E*). NREM slow-wave activity is a standard measure of sleep-intensity. These differences in SWA reveal underlying differences in sleep regulation between resilient and susceptible populations of mice. Collectively, our data demonstrate that sleep is both increased and reorganized after

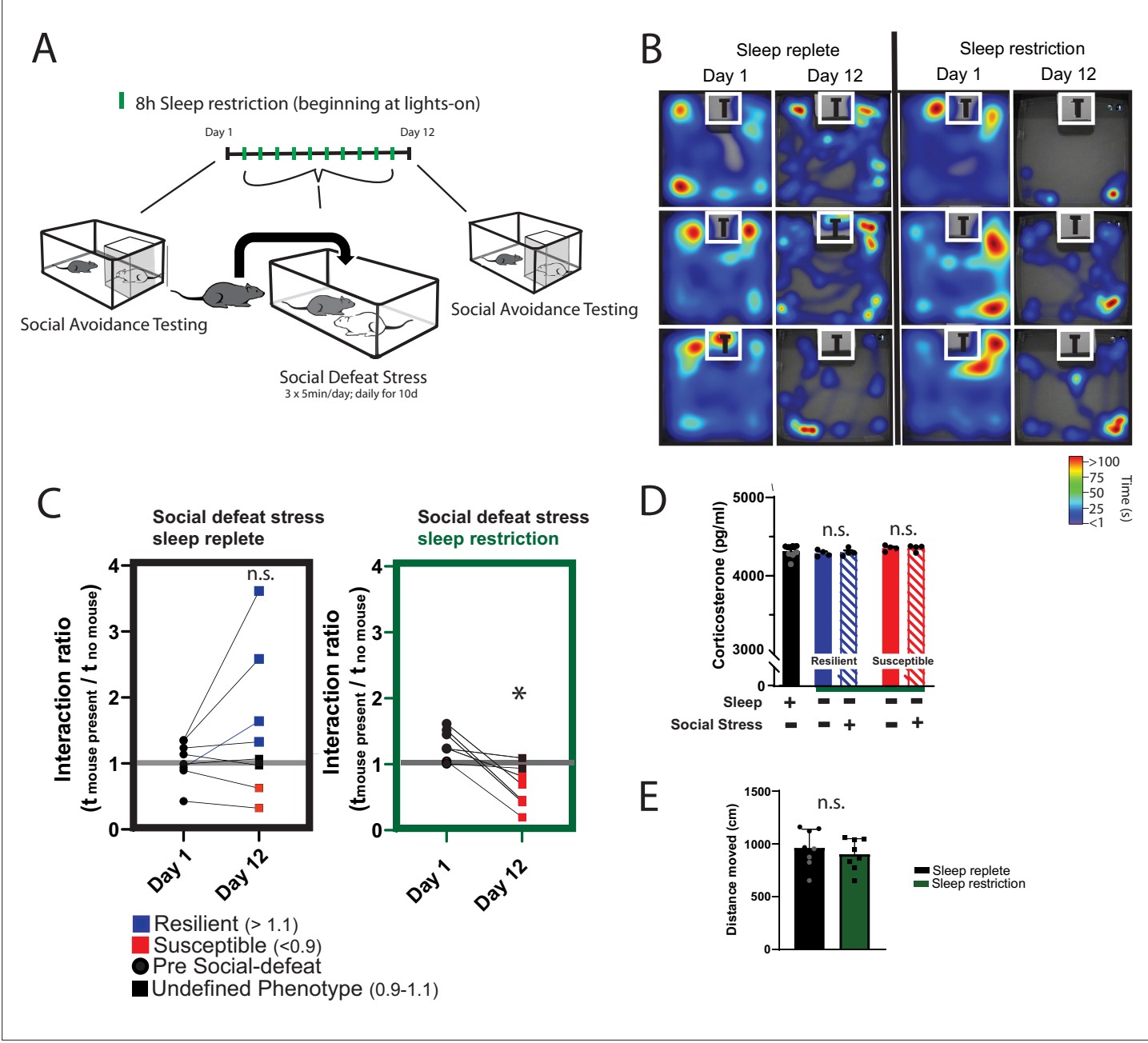

**Figure 1.** Daily sleep-restriction prevents resilience to social-defeat stress. One cohort of mice received sleep restriction (8 hr, beginning at light onset) on each day of social-defeat stress; a second cohort (sleep replete) received only social-defeat stress (A; 10 days total, sleep restriction procedure outlined in Methods). As expected in the sleep replete cohort, roughly equal amounts of resilience and susceptibility occurred after social-defeat stress (B, C). In contrast, no mouse that underwent sleep restriction was resilient to social-defeat stress (B, C). Neither the stress response (D; indicated by fecal corticosterone) nor the distance moved during behavioral testing (E) was significantly altered by sleep deprivation. (B) Heatmaps showing the time and location of representative mice during 3-min social-avoidance test both before and after 10 days of social-defeat stress. (C) Interaction ratios calculated from the heatmaps in B; black circles—pre-stress, red box—susceptible, blue box—resilient, black box—undefined; social avoidance was expressed as an interaction ratio based on the time (t) spent interacting (near white box) with a caged, novel CD1 target mouse vs. an empty cage (interaction ratio = $t_e$ / $t_{tl}$); sleep replete—Student's paired $t$, t(7)=1.54, p=0.17; sleep restriction—Student's paired $t$, t(6)=4.02, p=0.007; sleep replete—n=8, sleep restriction—n=7. (D) ANOVA, F(4,19)=1.12, p=0.37; n=8. (E) Student's $t$, t(14)=0.74, p=0.47. Data points represent mean ± SEM.

The online version of this article includes the following figure supplement(s) for figure 1:

**Figure supplement 1.** Aggressive encounters during 10 – days of social-defeat stress.

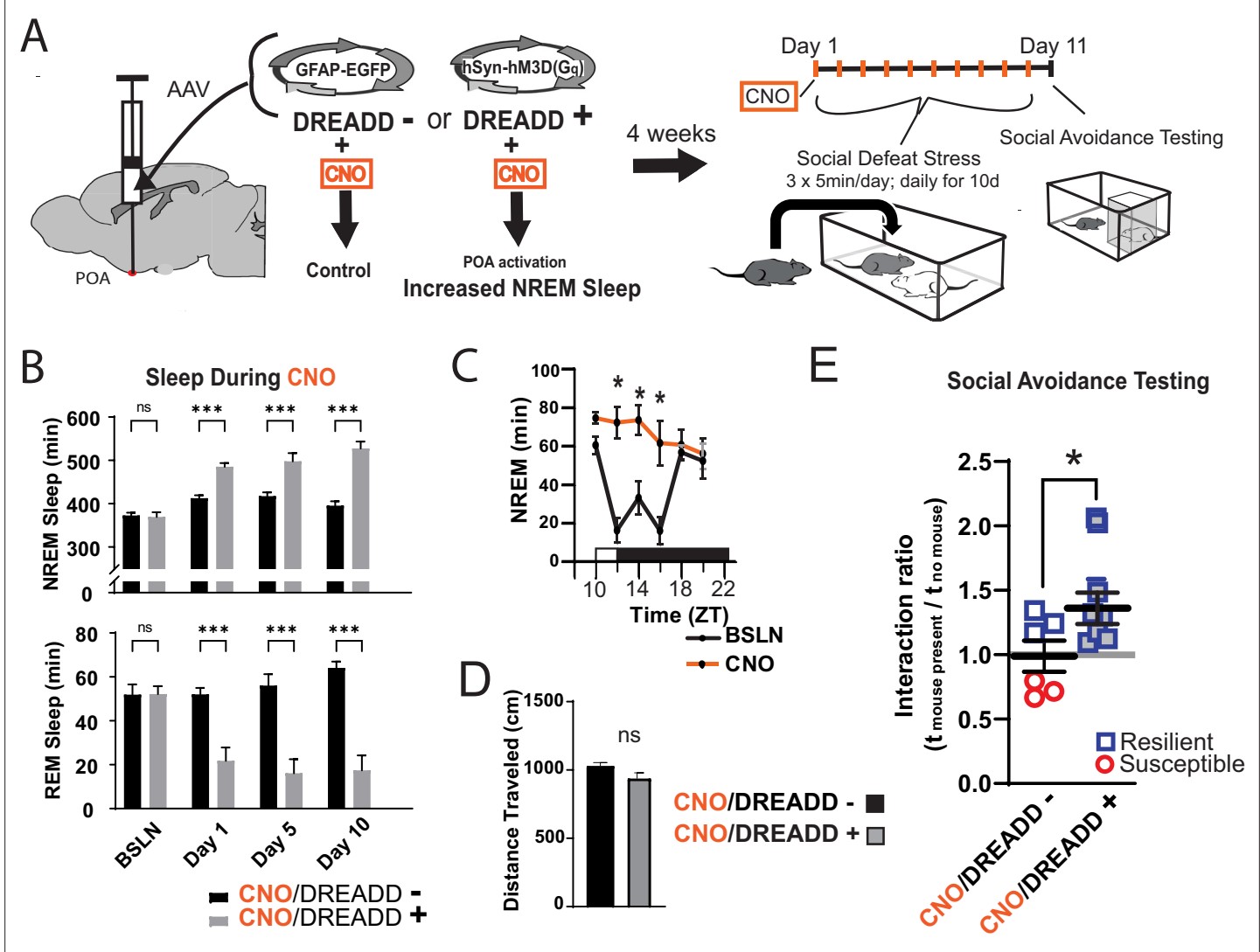

**Figure 2.** Increased sleep promotes resilience to social-defeat stress. Adeno-associated viral vectors (AAV2) encoding either an excitatory ($G_q$) designer receptor exclusively activated by designer drugs (DREADD), or enhanced green fluorescent protein (EGFP) as a control, were delivered to the preoptic area (POA) by intracranial microinjections. After four weeks, i.p. injections of clozapine N-oxide (CNO) were used to activate receptors expressed in the POA (**A**). In a validation study, chemogenetic activation of the POA significantly increased NREM sleep for six hours (compared to undisturbed sleep on the previous day) following a single injection of the agonist CNO at zeitgeber time 10 (C; ZT 10, ZT12=lights off). A separate cohort of mice expressing DREADD, or EGFP control, was exposed to 10 days of social-defeat stress (ZT12–13) with single, daily, i.p. injection of CNO at lights on (ZT 1–2; **A**). NREM sleep was significantly increased by daily injections of CNO in mice expressing the excitatory DREADD, but not in mice expressing the control DREADD (**B**). No mouse expressing the excitatory DREADD was susceptible to the effects of social-defeat stress (**E**). Mice expressing the control construct displayed both susceptible and resilient behavior as expected (**E**). The distance moved during behavioral testing was not significantly affected by POA activation (**D**). (**B**) Repeated measures ANOVA: NREM main effect of CNO—$F_{(1, 35)}$=83.37, p<0.0001; interaction effect—$F_{(3, 35)}$=13.43, p<0.0001; REM main effect of CNO—$F_{(1, 29)}$=72.92, p<0.0001; interaction effect—$F_{(3, 29)}$=10.5, p<0.0001; *, p≤0.001, Holms Sidak's multiple comparison; n=6, DREADD-; n=9 DREADD +. (**C**) Repeated measures ANOVA: main effect of CNO—$F_{(1, 8)}$=12.82, p=0.008; interaction effect—$F_{(5, 40)}$=8.04, p<0.0001; *, p≤0.001, Holms Sidak's multiple comparison. (**D**) Student's t, $t_{(11)}$=1.82, p=0.087. (**E**) Student's t, $t_{(11)}$=2.157, p=0.027. Data points represent mean ± SEM; *=p 0.05.

The online version of this article includes the following figure supplement(s) for figure 2:

**Figure supplement 1.** Histological verification of designer-receptor exclusively activated by designer-drugs (DREADD).

social-defeat stress exclusively in resilient mice and suggest that differences in sleep regulation may underlie these sleep changes.

We also investigated whether sleep-regulatory differences underlie the sleep responses of resilient and susceptible mice. Sleep is homeostatically regulated, as NREM sleep amount and intensity

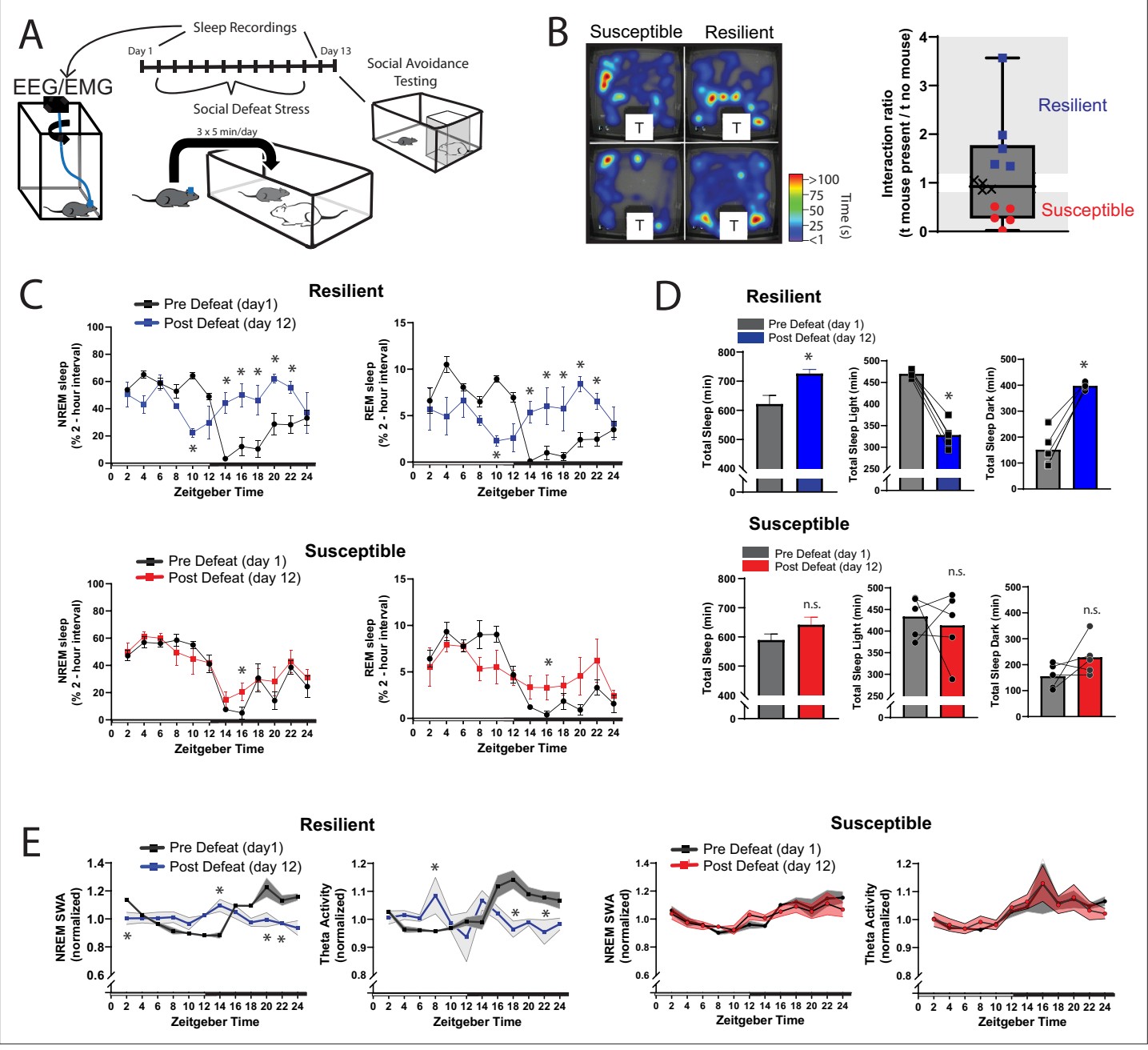

**Figure 3.** Sleep is reorganized only in mice resilient to social-defeat stress. 24 hr sleep recordings were performed both before and following 10 days of social-defeat stress (**A**). Sleep in animals identified as resilient (**B**) was significantly reorganized (**C**) and increased (**D**) following social-defeat stress. This change in resilient mice included a significant decrease in total sleep during the light period and increased total sleep during the dark period (**D**). Animals identified as susceptible (**B**) showed little change in sleep architecture or amount (**C, D**). Sleep changes in resilient animals included a flattening of the normal curve in sleep intensity (E; NREM slow-wave activity, SWA: power density 0.5–4 Hz). This change was accompanied by changes in higher frequencies during NREM sleep; theta activity was increased during the day and decreased during the night (**E**). In contrast, no significant change in either NREM slow-wave or theta activity was observed in susceptible animals (**E**). (**B**) Left, representative heatmaps of social avoidance testing; warmer colors indicate increased time; T=caged mouse; right, interaction ratios, X indicates interaction ratios between 0.9 and 1.1 that were excluded from sleep analysis. (**C**) Resilient, repeated measures ANOVA: NREM: main effect of time—$F_{(11, 88)}$=4.86, p=0.0001; main effect of day—$F_{(1, 8)}$=11.71, p=0.009; interaction—$F_{(11,88)}$=8.02, p=0.0001. REM: main effect of time—$F_{(11, 88)}$=3.17, p=0.0012; main effect of day—$F_{(1, 8)}$=0.95, p=0.358; interaction—$F_{(11,88)}$=6.7, p=0.0001. Susceptible, repeated measures ANOVA: NREM main effect of time—$F_{(11, 88)}$=8.581, p<0.0001; main effect of day—$F_{(1, 8)}$=2.925, p=0.1256; interaction—$F_{(11, 88)}$=1.429, p=0.1742; n=12. (**D**) Resilient, Student's paired t: total sleep—$t_{(5)}$=5.09, p=0.007; light—$t_{(5)}$=14.62, p=0.0001; dark—$t_{(5)}$=8.15, p=0.0012. (**E**) Resilient, SWA: main effect of time—$F_{(11, 84)}$=4.482, p<0.0001; main effect of susceptibility—$F_{(1, 8)}$=2.84, p=0.14; interaction—$F_{(11, 84)}$=6.47, p=0.0001. NREM theta activity: main effect of time—$F_{(11, 89)}$=1.97, p=0.042; main

*Figure 3 continued on next page*

*Figure 3 continued*

effect of susceptibility—$F_{(1, 89)}=5.04$, $p=0.027$; interaction—$F_{(11, 89)}=4.39$, $p<0.0001$. susceptible: main effect of time—$F_{(11, 84)}=6.31$, $p<0.0001$; main effect of susceptibility—$F_{(1, 8)}=0.64$, $p=0.45$; interaction—$F_{(11, 83)}=1.31$, $p=0.23$; *, $p\leq0.05$, Holms Sidak's multiple comparison. Data points represent mean ± SEM except for B which presents median, 25th to 75th percentiles and min/max values.

The online version of this article includes the following figure supplement(s) for figure 3:

**Figure supplement 1.** Sleep is not reorganized after exposure to a novel cage.

(slow-wave activity, SWA) are proportional to the duration of prior wakefulness (*Borbély et al., 1981*; *Dijk et al., 1987*). A standard method for investigating this homeostatic process involves restricting sleep and then measuring the resulting changes in NREM amount and intensity. To investigate sleep-regulatory differences that may underlie resilience to social-defeat stress, we used a sleep restriction paradigm both before and after exposure to social-defeat stress (*Figure 4A*). Prior to social-defeat stress, mice later identified as susceptible showed increased sleep-recovery (*Figure 4B, C*, top row), and increased NREM SWA (*Figure 3G*), after 6-hr of sleep restriction; mice later identified as resilient did not show these changes. This difference was not caused by the amount of sleep lost, as both behavioral phenotypes lost similar amounts of NREM sleep during sleep restriction (*Figure 4C*, bottom row). In addition, the increased recovery-response in susceptible mice remained when NREM sleep was normalized for each mouse (to the amount of sleep lost over 6-hr and 18 hr of recovery; *Figure 4C*, bottom row). After exposure to social stress, the overall patterns in NREM-sleep and NREM SWA were similar to pre social-defeat stress values (*Figure 4D and F*, bottom row). Notably, NREM SWA in susceptible mice was greater than resilient mice, both before (i.e. under baseline conditions) and after exposure to social-defeat stress (*Figure 4E & F,* top row). Both behavioral phenotypes also showed equivalent increases in REM-sleep and total sleep (*Figure 4D,* top row). These findings demonstrate that sleep and sleep-regulatory changes in resilient mice are not simply caused by social-defeat stress, instead, differences in sleep regulation exist prior to social-defeat stress and predict resilience. Collectively, our findings suggest that pre-existing differences in sleep regulation determine the sleep-response to social-defeat stress; these sleep responses, in turn, determine resilience.

The ventromedial prefrontal cortex (vmPFC; prelimbic, PrL and infralimbic, IL cortex) is important in regulating stress resilience (*Vialou et al., 2014*) and is also sensitive to the effects of sleep restriction (*Ehlen et al., 2013*; *Vyazovskiy et al., 2011*). The technique used for our EEG recordings (epidural screw electrodes) does not provide optimal spatial resolution. To better investigate local differences in sleep, we used local field potential recordings (LFP) from electrodes deep in the vmPFC. When compared to susceptible mice, mice later identified as resilientshowed a significant increase in baseline LFP power density in the slow-wave range (≤5 Hz) before social-defeat stress (*Figure 5A*). This SWA in baseline NREM sleep was found at both major vmPFC subregions targeted by our LFP-electrodes. The difference was most prominent in the PrL cortex and also detected in the IL cortex. Notably, no power differences were found in our EEG recordings (although a trend may be present ≤2 Hz; *Figure 5A*, *Figure 5—figure supplement 1*). These findings suggest that local differences in NREM slow-wave activity, between susceptible and resilient mice, exist in the vmPFC before exposure to social stress and can predict resilience.

Next, we looked at recovery responses to sleep restriction before exposure to social-defeat stress. Slow-wave energy was recovered in significantly less time for resilient animals, indicating a more efficient sleep-regulatory response (*Figure 5B*, *Figure 5—figure supplement 1*). Slow-wave energy returned to baseline levels immediately after sleep restriction in the PrL LFP and EEG of resilient mice, and within 2-hr in the IL LFP. Susceptible mice, in comparison, were not fully recovered for up to 8 hr (*Figure 5B*). We also examined the LFP during wake, both before and after sleep restriction. Low-frequency waveforms (2–6 Hz) increase in number and amplitude with sleep-pressure accumulation (*Ehlen et al., 2013*; *Thomas et al., 2000*). These waveforms are thought to represent local sleep-like events encroaching into wakefulness (i.e. sleepiness). Counting the occurrence of the largest amplitude 2–6 Hz waveforms ($I_{2-6}$) revealed that $I_{2-6}$, and thus waking sleep-pressure, was increased in resilient mice after sleep-restriction. Susceptible mice showed no significant change in $I_{2-6}$ within the awake LFP (*Figure 5C*). To further assess local differences in the vmPFC, we calculated phase coherence between each LFP electrode and the global EEG. We found phase coherence below 8 Hz increased in resilient mice during all conditions including: baseline, recovery from sleep restriction and after social-defeat stress; however, this difference was only found in the IL cortex (*Figure 6*).

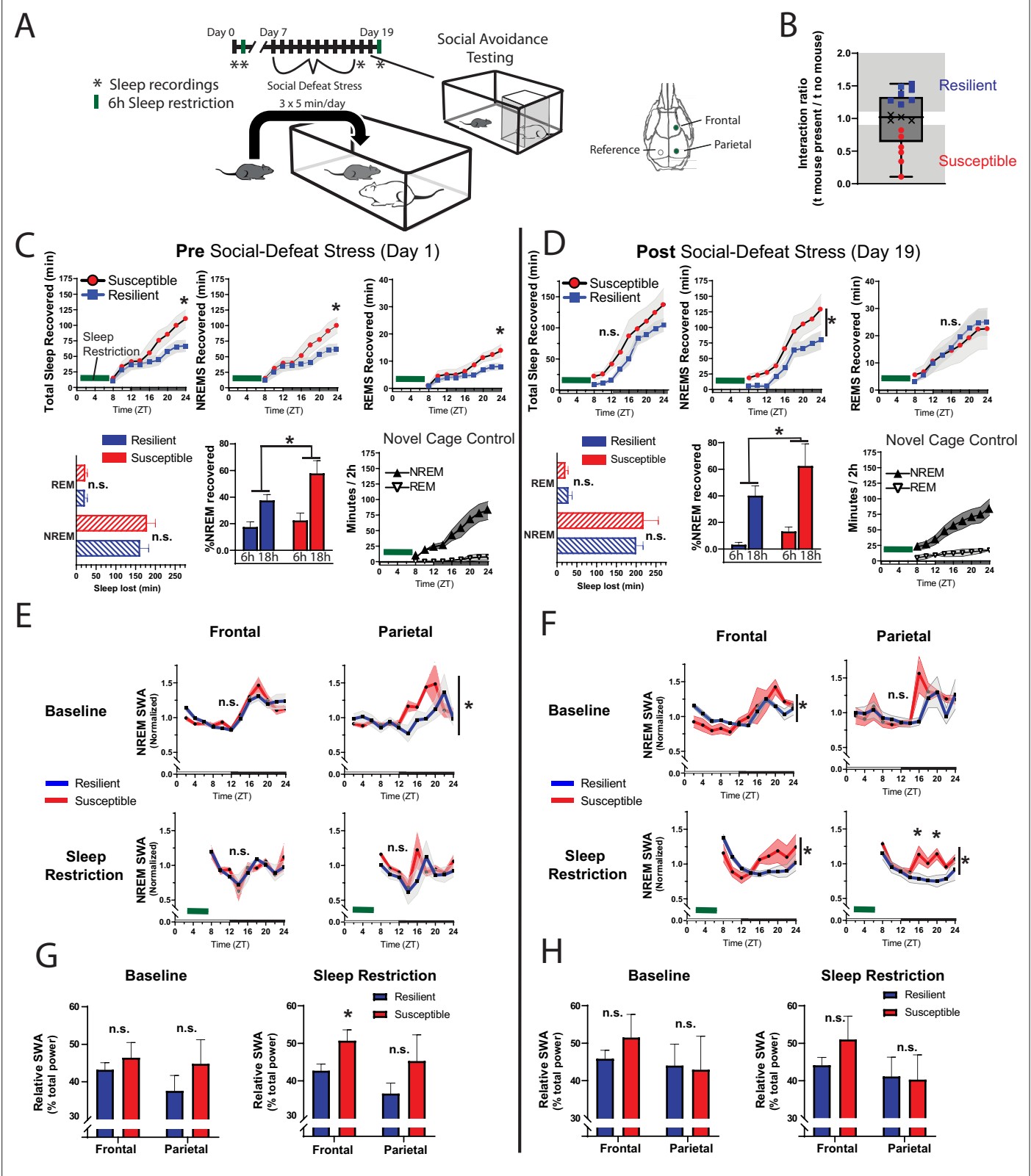

**Figure 4.** Differences in sleep regulation, prior to social-defeat stress exposure, predicts resilience to social-defeat stress. Sleep regulation involves a homeostatic process, as NREM sleep amount and intensity (slow-wave activity, SWA) are proportional to the duration of prior wakefulness. A standard method for investigating this sleep-regulatory process is restricting sleep and then measuring the resulting changes in NREM amount and intensity. Here, we used a six-hour sleep restriction paradigm both before and after 10 days of social-defeat stress to investigate the sleep-regulatory differences

*Figure 4 continued on next page*

*Figure 4 continued*

between resilient and susceptible mice (**A**). Prior to social-defeat stress exposure, mice later identified as susceptible showed increased sleep-recovery to six-hours of sleep restriction (B; C, top row) when compared to mice identified as resilient (B; C, top row). This pattern was present for both NREM and REM sleep (C, top row). Susceptible and resilient mice lost similar amounts of NREM and REM sleep during sleep-restriction (C, bottom row) and the increased recovery response for susceptible mice persisted when NREM sleep recovered was normalized to the amount of sleep lost (C, bottom). After social-defeat stress, susceptible animals continued to show increased NREM-sleep recovery, but not REM-sleep recovery (D, top row). Sleep intensity (NREM slow-wave activity, SWA) was higher in susceptible animals prior to social-defeat stress during both baseline and following sleep restriction (**E, G**). This increased sleep intensity in susceptible mice persisted and was more prominent after social-defeat stress (**F, H**). (**B**) X indicates interaction ratios between 0.9 and 1.1 that were excluded from sleep analysis. (**C**) Total sleep: repeated measures ANOVA main effect of time (MET)—$F_{(8, 80)}=53.78$, $p<0.0001$; main effect of susceptibility (MES)—$F_{(1, 10)}=2.05$, $p=0.18$; interaction (IST)—$F_{(8, 80)}=5.14$, $p<0.0001$. NREM sleep, MET—$F_{(8, 80)}=50.75$, $p<0.0001$; MES—$F_{(1, 10)}=1.92$, $p=0.19$; IST—$F_{(8, 80)}=4.77$, $p<0.0001$. REM sleep, MET—$F_{(8, 80)}=36.88$, $p<0.0001$; MES—$F_{(1, 10)}=2.73$, $p=0.13$; IST—$F_{(8, 80)}=3.32$, $p<0.0025$. Sleep lost, Student's paired t, NREM—$t(7)=0.99$, $p=0.36$; REM—$t(7)=0.92$, $p=0.37$. % recovered: MET—$F_{(8, 80)}=41.45$, $p<0.0001$; MES—$F_{(1, 10)}=1.58$, $p=0.24$; IST—$F_{(8, 80)}=3.27$, $p=0.0029$. (**D**) NREM sleep: MET—$F_{(8, 56)}=49.2$, $p<0.0001$; MES—$F_{(1, 7)}=3.96$, $p=0.049$; IST—$F_{(8, 56)}=1.86$, $p=0.23$. REM sleep: MET—$F_{(8, 56)}=24.58$, $p<0.0001$; MES—$F_{(1, 7)}=0.02$, $p=0.87$; IST—$F_{(8, 56)}=0.48$, $p=0.86$. (**E**) Baseline, parietal: MES—$F_{(1, 95)}=4.21$, $p=0.043$. (**F**) Frontal baseline: IST—$F_{(11, 75)}=2.13$, $p=0.028$; frontal sleep restriction, IST—$F_{(8, 63)}=2.13$, $p=0.028$; parietal sleep restriction, IST—$F_{(8, 63)}=2.84$, $p=0.009$. *=$p \leq 0.05$, Holms Sidak's multiple comparison; n=13. Data points represent mean ± SEM with the exception of panel B which presents median, 25th to 75th percentiles and min/max values.

The online version of this article includes the following figure supplement(s) for figure 4:

**Figure supplement 1.** Sleep fragmentation before and after social-defeat stress.

Importantly, the 24 hr average of these coherence values, in slow-wave frequencies, was significantly and positively correlated with social interaction ratios for the IL cortex, but not for the PrL cortex (*Figure 6B*). Overall, this phase-coherence data indicates increased functional connectivity (i.e. a consistent phase-relationship evidenced by high phase-coherence) between the IL cortex and the overall EEG, in the frequency bands associate with sleep, is correlated with resilience. These LFP studies further support our conclusion that local differences in sleep-regulatory processes within the vmPFC exist before stress and predict resilience.

## Discussion

We have applied behavioral, electrophysiological, and chemogenetic approaches to investigate a causal link between pre-existing sleep-differences and behavioral responses to stress. We demonstrate through sleep restriction that sleep is required for resilience to social-defeat stress; further, our POA-activation findings demonstrate that specifically increasing sleep amount increases resilience. Thus, sleep plays a determinative role in resilience to social-defeat stress. Furthermore, sleep recordings obtained before and after social-defeat stress reveal that increases in sleep are exclusive to resilient animals. Together, these data lend strong support for the essential role of sleep in conferring resilience.

Several previous studies have revealed that social stress in both mice and rats causes increased sleep intensity (*Meerlo et al., 1997*) notably, this increased sleep intensity also occurs in the 'winners' of social conflict (*Kamphuis et al., 2015*). *Meerlo et al., 1997* have suggested that the specific nature of preceding wakefulness as well as the duration of prior wakefulness is important in sleep responses (*Meerlo et al., 1997*). Thus, social-defeat stress may demand increased recovery sleep because it represents a more intense form of wakefulness (*Meerlo et al., 1997*). Indeed, the effect of waking-context on sleep regulation may extend beyond stress; recent findings indicate that repetitive tasks may represent less-intense wakefulness that, in turn, leads to less intense sleep (*Kamphuis et al., 2015*; *Milinski et al., 2021*). In this context, the altered sleep regulation of susceptible mice in our experiments may render them incapable of responding with increased sleep during or after social-defeat stress, whereas resilient animals adequately recover from this intense wakefulness. Although the ability of social-defeat stress to alter sleep regulation has been reported (*Henderson et al., 2017*; *Radwan et al., 2021b*), our studies demonstrate that differences in sleep regulation exist before exposure to social-defeat stress; however, it is important to consider that the mice were never truly stress naive as some manipulations prior to sleep recording (e.g. surgery) are stressful. The recent finding that sleep fragmentation, a potential indicator of altered sleep regulation, predicts stress susceptibility (*Radwan et al., 2020*; *Radwan et al., 2021a*) is consistent with this hypothesis of pre-existing sleep differences; fragmentation before stress exposure was also confirmed in our studies

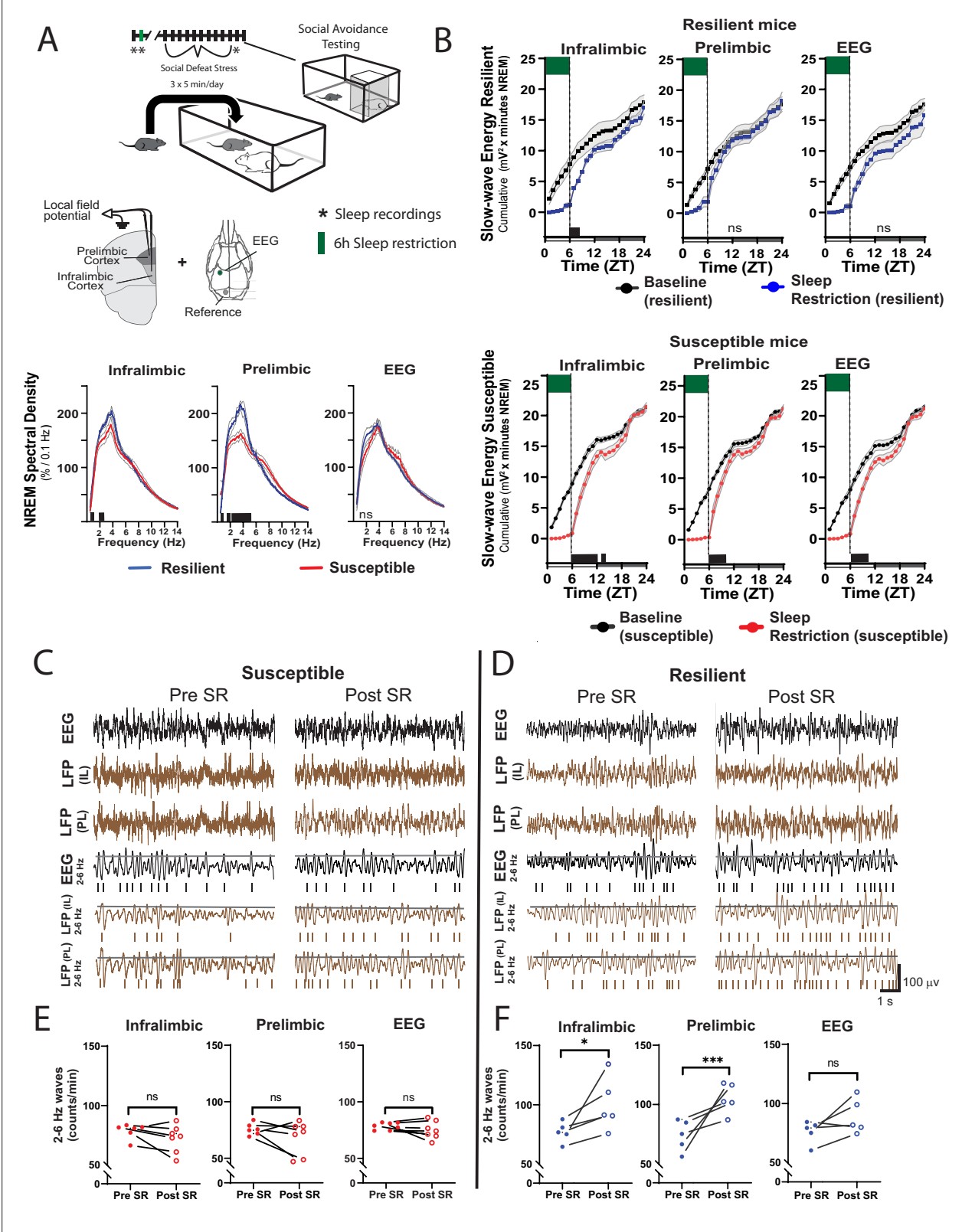

**Figure 5.** Sleep changes in the ventromedial prefrontal cortex (vmPFC) predict resilience to social-defeat stress. Local field potential (LFP) in the vmPFC and epidural electroencephalographic recordings (EEG; A, top row; black bars on x-axis=p ≤ 0.05) were simultaneously obtained from mice before ten-days of social-defeat stress. Twenty-four hr LFP/EEG recordings were immediately followed by 6-hr sleep restriction as in *Figure 4A*. Sleep regulation involves a homeostatic process, as NREM sleep amount and intensity (slow-wave activity, SWA) are proportional to the duration of prior wakefulness.

*Figure 5 continued on next page*

*Figure 5 continued*

A standard method for investigating this sleep-regulatory process is restricting sleep and then measuring the resulting changes in NREM amount and intensity. Here, we used a 6-hr sleep restriction paradigm before (no post-defeat restriction) 10 days of social-defeat stress to investigate the sleep-regulatory differences between resilient and susceptible mice. NREM sleep intensity (power density >4 Hz) in both the prelimbic and infralimbic LFP were significantly higher in mice later identified as resilient (vs. susceptible mice; A, bottom row); notably, these differences in sleep intensity were not evident in the EEG (A, bottom row). After 6-hr of sleep restriction resilient animals recovered at a faster rate than susceptible animals. This recovery is observed in cumulative NREM slow-wave energy (delta band = 0.5–4 Hz; $\text{energy} = \sum_{i=1}^{n} power_1 \times t_1$) and resilient mice took significantly less time to reach baseline levels (**B**). The incidence of 2–6 Hz waves in the waking EEG, a marker of sleep-pressure during waking, significantly increased in sleep restricted mice identified as resilient (**D**). This occurred in both LFP and EEG recordings and indicates a normal accumulation in sleep-pressure. Mice later identified as susceptible (**C**) did not show this increased wave-incidence, thus, indicating a lack of sleep-pressure accumulation. (**A**) Shaded areas are SEM; green boxes indicate sleep restriction; ANOVA; prelimbic interaction—$F$ (294, 2950)=4.55, p<0.0001; infralimbic main effect of susceptibility—F (294, 2950)=4.22, p<0.0001; EEG, interaction effect—$F$ (144, 1450)=0.9175, p=0.74; Holms Sidak's multiple comparison; n=12. (**B**) Shaded area on x-axis shows light–dark cycle. Repeated measures ANOVA interaction effect—susceptible mice: infralimbic LFP F(17, 187)=11.13, p<0.0001; prelimbic LFP F(17, 187)=5.71, p<0.0001; EEG F(17, 187)=6.75, p<0.0001; resilient mice: infralimbic LFP F(17, 136)=3.71, p<0.0001; black bars on x-axis=p < 0.05, Holms Sidak's multiple comparison. (**C, D**) Top rows—representative, raw EEG (black) and LFP (brown) recordings; middle rows—filtered EEG (black) and LFP (brown) signals (2–6 Hz) with threshold for counting identified (70th percentile, black line, see Materials and methods for details). (**E, F**) Lower plots—average wave-incidence for the 1 hr period immediately before and immediately after sleep restriction; Student's paired t, *, p=0.03; ***, p=0.0008.

The online version of this article includes the following figure supplement(s) for figure 5:

**Figure supplement 1.** Power density in resilient vs susceptible animals and sleep changes in the ventromedial prefrontal cortex.

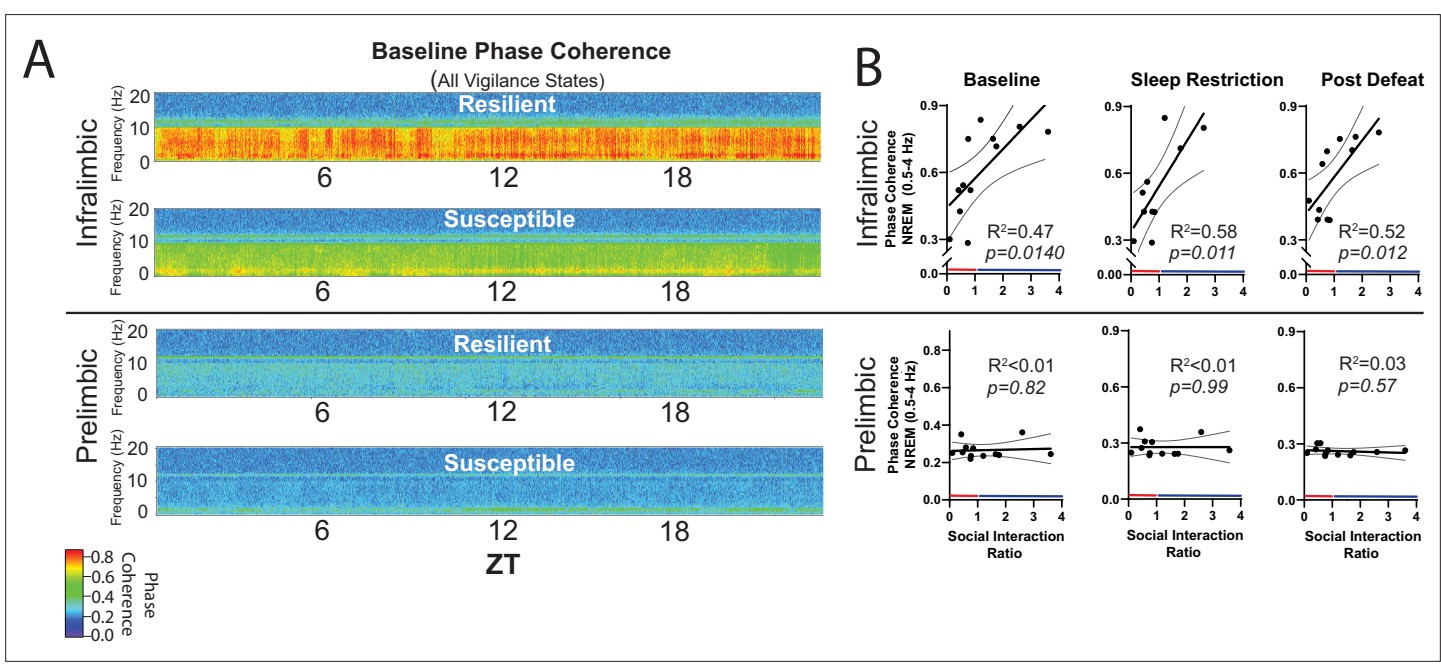

**Figure 6.** NREM coherence in the 0.5–4 Hz range predicts resilience to social-defeat stress. Local field potential (LFP) in the ventromedial prefrontal cortex and epidural electroencephalographic recordings (EEG) were simultaneously obtained from mice before and after 10-days of social-defeat stress (see *Figure 5*). Coherence below 8 Hz between the infralimbic (IL) LFP and EEG was significantly increased across the 24 hr day in undisturbed, resilient mice (compared to susceptible; A, top; warmer colors represent increased coherence). This increased coherence with the EEG below 10 Hz was also visible in the prelimbic (PrL) LFP, but this effect was reduced and not significant (A, bottom; compared to susceptible). A similar pattern of increased coherence in resilient animals was observed after 6 hr of sleep restriction and after 10 days of social-defeat stress (data not shown). Notably, IL-coherence (averaged over 24 hr) predicted social interaction ratios (B, top) both before (B, left) and after (B, middle) social-defeat stress and during recovery from six-hours of sleep restriction (**B**). These correlations were not observed for the PrL cortex (B, bottom). (**A**) Heatmaps of coherence over 0.1 Hz intervals in 10 min bins; IL, repeated measures ANOVA: main effect of susceptibility—F(1, 10)=19.81, p=0.0024; PrL, main effect of susceptibility—F(1, 10)=0.188, p=0.09 n.s; n=12. (**B**) Least-squares regression line with 95% confidence interval and goodness of fit (R²); colors represent resilience (blue) or susceptibility (red) based on interaction ratio; non-zero slope of least-squares regression line, baseline—F(1, 10)=8.87, sleep restriction—F(1, 10)=10.99, post social defeat—F(1, 10)=9.83; p values provided in plots.

(*Figure 4—figure supplement 1B*; longer NREM bout durations in resilient mice). Thus, the sleep changes, and ultimate behavioral outcomes resulting from social-defeat stress, are likely the result of pre-existing differences in sleep regulation. In this context, the intense waking experience of social stress, interacting with differences in sleep regulation between susceptible and resilient mice, may allow resilient animals to recover sleep and leave susceptible animals in a stress-vulnerable, perpetually sleep-deprived state.

Susceptible mice appear to be sleep deprived in that they show markers of insufficient sleep after 6 hr of sleep restriction. The deprivation was indicated by increased recovery sleep, both before and after social-defeat stress (*Figure 4C and D*); as well as a delayed recovery of slow wave energy (*Figure 5B*). Insufficient sleep causes both a reduced ability to cope with stress and negative effects on mood (for a review see *Goldstein and Walker, 2014*). Functional magnetic resonance imaging studies reveal that sleep deprivation leads to decreased functional connectivity between the ventromedial prefrontal cortex (vmPFC) and amygdala; furthermore, this decreased functional connectivity is associated with decreased mood (*Killgore, 2013*; *Motomura et al., 2017*; *Yoo et al., 2007*; *Drummond et al., 1999*). This same brain region, the vmPFC, is known to mediate resilience to social-defeat stress (*Vialou et al., 2014*; *Lehmann and Herkenham, 2011*; *Kumar et al., 2014*; *Hultman et al., 2018*). We reasoned that the sleep deprived state of susceptible mice would lead to vmPFC dysfunction; thus, leading to decreased connectivity and an inability to inhibit the limbic circuits responsible for regulating behavioral responses to social stress. As predicted, we found major differences between the vmPFC of susceptible and resilient mice. Our LFP recordings reveal that baseline slow wave activity and recovery from sleep deprivation are preferentially enhanced in the vmPFC of resilient animals—before exposure to stress (*Figure 5A and B*). Wave incidence after sleep restriction, a marker of waking sleep pressure, is also significantly increased in resilient mice (*Figure 5C–F*). Together, the enhanced buildup of sleep pressure and increased homeostatic response led to faster recovery from sleep restriction in the vmPFC of resilient mice, thus supporting the hypothesis that NREM-related changes in vmPFC are associated with resilience.

In the present study, we used activation of the POA as a method to increase sleep. The POA has a well-established role in the promotion of sleep and multiple cell groups and neuronal subtypes within this region are involved in initiating sleep (*Sherin et al., 1996*; *Liou et al., 1990*; *Chung et al., 2017*). Other physiological responses are also regulated by the POA including exploratory and sexual behavior, shivering thermogenesis and body temperature (*Conceição et al., 2019*; *Tsuneoka and Funato, 2021*; *Kroeger et al., 2018*). A recent chemogenetic study activated galanin-expressing neurons in the POA and reported findings similar to those reported here (increased NREM and decreased REM sleep), and a decrease in body temperature (*Kroeger et al., 2018*). We did not observe behavioral effects other than sleep and our method was different in that it activated all neurons in the region; however, we cannot completely rule out the occurrence of non-specific effects. Furthermore, it is not clear from our studies if either clozapine N-oxide (CNO) or G$_q$ DREADD +CNO (POA activation) influences social-avoidance behavior in the absence of stress. To strengthen our findings, we conducted a detailed histological examination of this region to verify DREADD expression. All mice in the study expressed DREADD in regions of the POA known to initiate sleep (*Figure 2—figure supplement 1*) and all responded with increased NREM sleep (*Figure 2*). Nevertheless, when considered in the context of our other results, especially our finding that sleep restriction decreases resilience (i.e. sleep restriction has the opposite effect of POA activation), our studies provide strong evidence that changes in sleep mediate the changes we observed in resilience.

Sleep regulation in the cortex is known to occur at a local level (*Vyazovskiy et al., 2011*; *Steriade et al., 2001*; *Miyamoto et al., 2003*); areas that are more active during awake periods are known to have increased sleep intensity during subsequent sleep episodes (*Pigarev et al., 1997*; *Dang-Vu, 2012*). To find if local sleep-differences contribute to resilience, we examined sleep-related changes in two major subdivisions of the vmPFC—the infralimbic (IL) and prelimbic (PrL) cortex. Phase-coherence in the IL cortex (0.5–4 Hz, with the epidural EEG) was significantly and positively correlated with social-interaction ratios (*Figure 6*). This positive correlation increased after sleep restriction and persisted after social stress. No such positive correlation was found in the PrL cortex. Increased phase-coherence suggests an increase in functional connectivity between the IL cortex with the global EEG (in NREM frequency ranges) and suggests that this increased functional connectivity between these brain regions predicts resilience. Phase coherence does not indicate directionality of this functional connectivity;

however, it does provide evidence that local differences in sleep are correlated with maladaptive behavioral responses to stress. In future studies, it will be important to determine if causation can be demonstrated for these relationships. Other subregional differences included enhanced SWA (5 A, 0.5–4 Hz range) and slow-wave-energy recovery (*Figure 5B*) in the PrL cortex. Collectively, these findings suggest that the sleep-regulatory differences predicting resilience can be further localized to specific subregions of the vmPFC. Both the IL and PrL cortex have a demonstrated importance in resilience to social-defeat stress (*Vialou et al., 2014*; *Lehmann and Herkenham, 2011*; *Covington et al., 2010*; *Dulka et al., 2020*) thus, the significance of the observed differences between subregions of the vmPFC (e.g. coherence, power density and slow-wave energy) is not yet clear. Nevertheless, the data demonstrate a clear positive correlation between NREM sleep in the vmPFC and resilience.

Our findings indicate a major role for NREM sleep in mediating resilience to social-defeat stress based on several lines of evidence. First, our sleep deprivation paradigm decreased both NREM and REM sleep, whereas activation of the POA preferentially increased NREM sleep while REM sleep was reduced. Because REM was decreased in both conditions, with opposite effects on social interaction, REM sleep is unlikely to have large effects on resilience. Furthermore, evidence of altered NREM sleep regulation was most prominent in the vmPFC of resilient mice. In this area, the frequency bands that predominate in NREM sleep were significantly higher and NREM responses to sleep deprivation were significantly enhanced in resilient mice (*Figures 5 and 6*). Together, these findings strongly implicate NREM sleep in mediating changes in resilience, but do not rule out the involvement of REM sleep.

The effects of stress on behavior, and sleep-responses to stress, vary with sex (*Paul et al., 2009*; *Paul et al., 2006*; *Yohn et al., 2019b*) thus, it is not possible to predict how our findings relate to females. Female mice are not territorial; therefore, social-defeat stress in females requires alternative defeat-paradigms. These female-defeat paradigms were only recently developed and reported (*Yohn et al., 2019b*; *Yohn et al., 2019a*), which prevented us from considering sex as a biological variable in the present study. Studies in females will be critical to understanding the role of NREM sleep in resilience and are currently underway in our lab.

The present studies show that sleep is an active response to social stress that serves to promote resilience, thus demonstrating a clear causal link between insufficient sleep and maladaptive behaviors. Further, our findings in the vmPFC reveal local changes in sleep that may not be visible in the global EEG. This induced sleep response is dependent on inter-individual variability in sleep regulation and, if sufficient sleep is obtained, likely serves to mitigate the well-established negative effects of sleep loss on CNS function—including cognition and emotion (*Krause et al., 2017*; *Ritland et al., 2019*; *Simonelli et al., 2019*). In addition, the newly demonstrated ability of sleep manipulations to alter behavioral responses to stress offers new possibilities for this mouse model of resilience. This model can be used to investigate the specific mechanisms by which sleep alters stress-induced changes in brain physiology and behavior.

## Materials and methods

### Key resources table

| Reagent type (species) or resource | Designation | Source or reference | Identifiers | Additional information |
|---|---|---|---|---|
| Strain, strain background (*Mus musculus*, male) | C57BL/6 J | Jackson Labs, Bar Harbor, ME. | Stock #: 000664 | |
| Strain, strain background (*Mus musculus*, male) | CD-1; retired breeders | Charles Rivers Laboratories, Willington, MA. | 0022CD1 | Used as aggressors in social defeat |
| Recombinant DNA reagent | Control Vehicle, DREADD-, pAAV-hSyn-EGFP | Addgene, Watertown, MA. | Plasmid #50465 | |
| Recombinant DNA reagent | Excitatory DREADD, DREADD +, pAAV-hSyn-hM3D(Gq)-mCherry | Addgene, Watertown, MA. | Plasmid #50474 | |
| Commercial assay or kit | Corticosterone ELISA Kit | Cayman Chemical, Ann Arbor, MI. | Item No. 501320 | |

*Continued on next page*

*Continued*

| Reagent type (species) or resource | Designation | Source or reference | Identifiers | Additional information |
|---|---|---|---|---|
| Chemical compound, drug | Clozapine N Oxide; CNO | Hello Bio Princeton, NJ. | Cat# HB6149 | Dosage: 2 mg/kg |
| Software, algorithm | Sirenia Acquisition | Pinnacle Technology Inc, Lawrence, KS. | Version 1.8.3 | |
| Software, algorithm | Sirenia Sleep | Pinnacle Technology Inc Lawrence, KS. | Version 1.8.3 | |
| Software, algorithm | Sleep Deprivation System | Pinnacle Technology Inc, Lawrence, KS. | Cat. #: 9000-K5-S | |
| Software, algorithm | Igor Pro 8 software 64-bit | WaveMetrics, Inc, Lake Oswego, OR. | Version 8.04 | FilterIIR, DSPPeriodogram and custom scripts. |
| Software, algorithm, | Noldus Ethovision XT | Noldus Information Technology, Leesburg, VA. | Version 14 | Video tracking during social avoidance test. |
| Software, algorithm | GraphPad Prism | GraphPad Software, San Deigo CA. | Version 7.00 | |
| Other | Prefabricated Electroencephalographic Implant; EEG; electromyograph; EMG | Pinnacle Technologies Inc, Lawrence, KS. | Cat. #: 8201-SS; 8431 | Materials for surgery (Surgery: EEG and LFP electrodes section). |
| Other | Stainless steel screw electrodes | Pinnacle Technology Inc Lawrence, KS. | Cat. #: 8209, 8212, 8403 | Materials for surgery(Surgery: EEG and LFP electrodes section). |
| Other | Silver Epoxy | Pinnacle Technology Inc Lawrence, KS. | Cat. #:8226 | Materials for surgery(Surgery: EEG and LFP electrodes section). |

## Animals

Male, C57BL/6 J mice (Jackson Laboratory, Bar Harbor, ME, USA; 000664) were seven-weeks old at the start of all studies. CD-1 retired male breeders (Charles Rivers, age 3–6 months upon arrival) were used as aggressors. All mice were singly housed on shaved, pine bedding upon arrival, maintained on a 12:12 L:D lighting cycle for the remainder of the study and randomly assigned to treatment groups. Food and water were available ad libitum. All procedures involving animals received prior approval from the Morehouse School of Medicine Institutional Animal Care and Use Committee (approved protocol 21–02).

## Social defeat and social avoidance test

Mice were exposed to three daily, 5-min, social defeat sessions (separated by 5-min breaks) for 10 days (during the first 2 hr of the dark period; ZT 12–14). The C57BL/6 J mice to be defeated were placed into the home cage of a trained CD-1 mouse (aggressor). Each 5-min session was with a novel aggressor, and no defeated mouse encountered an aggressor more than twice over the 10 days of social-defeat stress. This social-defeat stress method varies from a popular protocol described by *Golden et al., 2011* our method adds one additional five minute social-defeat session. Importantly, after defeat, animals in our study were returned to their home cage; they did not spend the remainder of each day in a divided cage with the aggressor as described in *Golden et al., 2011*. Training of aggressor mice was conducted on each of the 3 days prior to testing. Training consisted of the same three, five-minute sessions, but was performed with a C57BL/6 J training mouse not used in experiments. Only aggressor mice that displayed at least five incidences of aggression for 2 consecutive days were used in the study. Social defeat sessions were continuously monitored, and the mice were separated for 10 s if excessive aggression (>15 seconds of continuous aggression) was observed. No significant wounding occurred during any of our defeat sessions. Small bite wounds were found on occasion, and they were treated with betadine after the defeat sessions ended. Dim red light (<5 lux) was used for procedures performed during the mouse's dark period. Novel cage controls were performed with the exact same procedures as social-defeat stress, but with the aggressor mouse moved to a holding cage.

Social avoidance testing was conducted in a 30x30 cm arena with a caged (9X9 cm), novel, CD1 mouse positioned against the midpoint of one arena wall. Each mouse to be tested was placed in the arena for two consecutive sessions of 3 min. During the first session, the cage was empty; during the second session, the novel CD1 mouse was present. Position in the arena was monitored with video tracking (Noldus Ethovision XT, Leesburg, VA, USA). The arena was cleaned thoroughly between each test. Social interaction ratio, *int,* was calculated as:

$$int = \frac{t_f}{t_e}$$

where $t_e$ = time within 15 cm of the empty cage in the first session, $t_f$ = time within 15 cm of the caged, novel CD1 mouse in the second session; ratio >1.1 = resilient, ratio <0.9 = susceptible. Mice with interaction ratios between 0.9 and 1.1 (n=11) were excluded from analysis in the experiments of *Figures 3–6* as follows: *Figure 3*, four mice excluded; *Figure 4*, four mice excluded; *Figures 5 and 6*, three mice excluded.

## Sleep restriction

Sleep restriction was accomplished by one of two methods and conducted for one of two durations, depending on the goals of the experiment. Sleep regulation involves a homeostatic process, as NREM sleep amount and intensity (slow-wave activity, SWA) are proportional to the duration of prior wakefulness. A standard method for investigating this sleep-regulatory process is restricting sleep and then measuring the resulting changes in NREM amount and intensity. We used a 6-hr sleep restriction paradigm before and after 10 days of social-defeat stress to investigate the sleep-regulatory differences between resilient and susceptible mice. In *Figures 4–6*, 6-hr sleep restriction was conducted once to monitor this sleep-regulatory process during the 18-hr period following sleep restriction. In *Figure 1*, sleep restriction was used as an intervention to reduce sleep during social-defeat stress and determine how this sleep restriction altered behavioral-responses to social stress. This sleep restriction was performed for the first 8 hr of the light phase (ZT 0–8) daily for each of the 10 days of social-defeat stress. Mice in this study were continuously monitored during this sleep restriction by trained observers.

One of two methods of sleep restriction were used depending on the number of mice and duration of the study. Sleep restriction in *Figure 3* was conducted by hand and performed by trained observers. Mice were kept awake during the first 6 hs of the light phase (ZT 0–6) once by gentle handling (introduction of novel objects into the cage, tapping on the cage and when necessary, delicate touching) and allowed an 18 hr recovery opportunity (ZT 6–0). In *Figures 1, 5 and 6* sleep deprivation was accomplished using an automated system (Pinnacle Technology, Inc Lawrence, KS) which maintained wakefulness by means of a slowly rotating bar in the cage bottom. The bar direction was set to change randomly every 10–20 s. In *Figures 5 and 6*, mice were kept awake for the first 6 hs of the light phase (ZT 0–6) once and allowed an 18 hr recovery opportunity (ZT 6–24). In *Figure 1*, sleep restriction was used as an intervention to reduce sleep during social-defeat stress and was performed for the first 8 hr of the light phase (ZT 0–8) daily for each of the 10 days of social-defeat stress. Sleep restriction was always carried out in the home cage of the mouse and food and water were available ad libitum.

## Corticosterone

Cages were changed 24 hr prior to each sample collection at ZT 12 (lights off). All feces from the cage were collected exactly 24 hr after this cage change and immediately frozen (–80 °C) until assay. Control samples were taken during the 24 hr period immediately preceding the first day of treatment. As second sample was taken 6 days later, during the 24 hr period that followed exposure to 5 consecutive days of social-defeat stress or 5 consecutive days of both social-defeat stress and sleep deprivation. Just prior to assay, the fecal sample was homogenized and 50 mg was removed for analysis. Sample preparation and analysis was done using the Cayman Corticosterone ELISA kit (Ann Arbor, Michigan), according to the kit -booklet instructions for extraction from feces and analysis. Each condition was run in duplicate, an average of these values was used in analysis.

## Surgery: EEG and LFP electrodes

Electroencephalography (EEG): EEG and Electromyography (EMG) electrodes were implanted in isoflurane (1.5–3%) anesthetized mice. Carprofen was given post operatively for 2 days. A prefabricated

head mount (Pinnacle Technology Inc, Lawrence, KS) was used to position three stainless-steel epidural screw electrodes. The first electrode (frontal—located over the frontal cortex) was placed 1.5 mm anterior to bregma and 1.5 mm lateral to the central suture, whereas the second two electrodes (inter-parietal—located over the visual cortex and common reference) were placed 2.5 mm posterior to bregma and 1.5 mm on either side of the central suture. The resulting two leads (frontal–interparietal and interparietal–interparietal) were referenced contralaterally. A fourth screw served as a ground. Electrical continuity between the screw electrode and head mount was aided by silver epoxy. EMG activity was monitored using stainless-steel Teflon-coated wires that were inserted bilaterally into the nuchal muscle. The head mount (integrated 2×3 pin socket array) was secured to the skull with dental acrylic. Mice were allowed to recover for at least 14 days before sleep recording.

Local field potential (LFP): LFP, EEG and EMG electrodes were identical to the EEG surgery described above with the following exceptions. A custom-made implant, consisting of two unipolar tungsten electrodes permanently attached to a 2x4 pin socket (Pinnacle Technology Inc, Lawrence, KS), was lowered through a craniotomy with the aid of a stereotaxic apparatus (David Kopff Instruments, Tujunga, CA). The implant was positioned so that the electrodes tips were in the infralimbic (anterior posterior [AP]:+1.9, medial lateral [ML]: –0.4, dorsal ventral [DV]: –3.1; coordinates relative to bregma and midsagittal suture) and prelimbic (AP:+1.9, ML:+0.4, DV: 4.45) cortex. Four epidural, stainless-steel screw-electrodes (Pinnacle Technology Inc, Lawrence, KS) were then positioned on the skull as follows: one recording electrode and LFP reference were placed contralaterally over the frontal cortex, two electrodes over the cerebellum served as an EEG reference and ground. Wire leads attached to each screw were then soldered to output pins on the implant. The implant and leads were covered and secured with dental acrylic.

## Excitatory DREADD

Bilateral injections into the preoptic area (POA) were performed in isoflurane (1.5–3%) anesthetized mice. A 0.5 μl microliter syringe needle (Hamilton, Reno, NV) was positioned in the POA, through a craniotomy made in the skull, with the aid of a stereotaxic apparatus (AP:+0.20, ML:±0.55 DV: –5.65). 200 nl of adeno-associated virus 2 (AAV2) containing either control construct (pAAV-hSyn-EGFP; plasmid #50465; Addgene, Watertown, MA) or excitatory DREADD (pAAV-hSyn-hM3D(Gq)-mCherry; plasmid #50474; Addgene, Watertown, MA) was delivered to the POA at 1 nl/s using a motorized syringe pump (World Precision Instruments, Sarasota, FL). The syringe needle was left in place for 10 min before removal. Mice were given 14 days of recovery before EEG implant surgery. Carprofen was given post operatively for 2 days.

## Clozapine N-Oxide (CNO)

Clozapine N-oxide dihydrochloride (CNO; 2 mg/kg; Cat# HB6149; HelloBio, Princeton, NJ) was diluted in in lactated ringers and delivered by intraperitoneal (IP) injections on each day of social defeat, where indicated (*Figure 2A*), between ZT1 and ZT2 (ZT12=lights off, early light phase). Mice expressing both control construct and excitatory DREADD received CNO. For validation studies (*Figure 2C*), single IP injections were delivered at ZT 10.

## EEG/LFP recording/scoring

One week after surgery, mice were moved to an open-top sleep-recording cage and connected to a lightweight tether attached to a low-resistance commutator mounted over the cage (Pinnacle Technologies, Lawrence KS). This enabled complete freedom of movement throughout the cage. Except for the recording tether, conditions in the recording chamber were identical to those in the home cage. Mice were allowed a minimum of seven additional days to acclimate to the tether and recording chamber. Data acquisition was performed on personal computers running Sirenia Acquisition software (Pinnacle Technologies). EEG signals were low-pass filtered with a 30 Hz cutoff and collected continuously at a sampling rate of 400 Hz. LFP signals, and EEG signals collected simultaneously with LFP, were low-pass filtered with a 1000 Hz cutoff and collected continuously at a sampling rate of 2 kHz. In most cases, sleep recordings were conducted in blocks of 8 mice including both treatments and controls. For classification of waveforms, these EEG signals were low-pass filtered offline at 30 Hz. After collection, all waveforms were classified by a trained observer (using both EEG leads and EMG; in 10 s. epochs) as wake (low-voltage, high-frequency EEG; high-amplitude EMG), NREM

sleep (high-voltage, mixed-frequency EEG; low- amplitude EMG) or rapid eye movement (REM) sleep (low-voltage EEG with a predominance of theta activity [6–10 Hz]; very low amplitude EMG). In all studies, individuals performing sleep-stage classification were blind to the experimental conditions and behavioral phenotypes until final analysis. EEG epochs determined to have artifact (interference caused by scratching, movement, eating, or drinking) were excluded from analysis. Artifact comprised less than 5% of all recordings used for analysis.

## Data analysis

Wave incidence analysis has been described previously (*Ehlen et al., 2013*; *Ehlen et al., 2015*). Briefly, analysis was performed using custom written functions in IGOR Pro 8 (WaveMetrics Inc, Lake Oswego, OR). Raw EEG and LFP signals were band pass-filtered in the frequency range indicated using a Butterworth fourth-order band-pass filter (IGOR Pro routine FilterIIR; WaveMetrics Inc, Lake Oswego, OR). Peaks in the filtered data were detected as negative deflections between two zero crossings. The upper 30% of peak amplitudes that occurred in epochs identified as wake were then counted and expressed as peaks per minute (wave incidence). The wave incidence data were binned for graphing and statistical analysis.

Power spectral analysis was accomplished by applying a fast Fourier transform (FFT, 0.1 Hz frequency resolution) to EEG or LFP recordings. Where indicated, spectral power within a frequency band was normalized to 24 hr baseline values for each animal or expressed as a percentage of total power (0.5–30 Hz) for each animal. Slow-wave energy, *energy*, was calculated using the 0.5–4 Hz (delta) frequency range as follows:

$$\text{energy} = \sum_{i=1}^{n} power_1 \times t_1$$

Phase coherence is calculated utilizing the Igor Pro DSPPeriodogram function which can calculate the degree of coherence between the input of two sources, in this case LFP and/or EEG records stored in the same EDF file and on the same time base. According to the Igor Pro literature, the coherence, *coh*, is given by:

$$coh = \frac{\sum_{i=0}^{M} F(x_i) \left[ F(y_i) \right]^*}{\sqrt{\sum_{i=0}^{M} F(x_i) \left[ F(x_i) \right]^* \sum_{i=0}^{M} F(y_i) \left[ F(y_i) \right]^*}}$$

where $F(x_i)$ and $F(y_i)$ are the Fourier transforms of the first and second EEG or LFP data chunks, *i*, for the same time period. The complex conjugate of the Fourier transform is symbolized by the standard notation of [ ]*. The data were recorded at 2 kHz. The data in the coherence plot are analyzed in 10 min bins made from successive chunks (i) of 2 s each and averaged according to the above equation (*M*=300 times). Igor Pro also applied a Hanning window to the data to remove edge effects. Frequency data resolution was 0.5 Hz. After Fourier analysis, the real and imaginary parts of each point of coherence on the frequency spectrum, *coh*, were squared and summed to yield a real number values which is the power. This is displayed between 0.5 and 20 Hz (each column on the coherence plot in *Figure 6*).

Sleep data were analyzed using one-way or two-way analysis of variance (ANOVA) with repeated-measures when appropriate. Student's t was used where indicated and for all tests significance was defined as *P*<0.05. Post hoc analysis was conducted using the Holm-Sidak method which adjusts α to maintain the family-wise error-rate at 0.05. Sample sizes (biological replicates) for each experiment are indicated in the figure legends. An appropriate sample size of 6 was predicted with Type I error rate of 0.05 and Type II error rate of 0.2. Standard deviation and mean difference were estimated as 14.6 and 25 min, respectively, based on previous recordings of C57BL/6 J mice obtained in our lab.

## Histology

Under deep isoflurane anesthesia, mice were perfused transcardially with 50 mL of cold 1M phosphate buffer saline (PBS) followed by 50 mL 0.4% paraformaldehyde (PFA). Brains were removed and post-fixed in 0.4% PFA for two days and then transferred to 1M PBS until sectioning. Coronal cryostat sections (25 μm) we transferred to glass slides and air dried. For LFP experiments, sections were stained with cresyl-violet and electrode locations were verified using light microscopy. Brain sections from DREADD experiments were mounted with DAPI-containing mounting medium (nucleic-acid

counter-stain; Fluoromount-G, Invitrogen); the presence and location of DREADD expressing cells were verified with laser scanning confocal microscopy (*Figure 2—figure supplement 1*; LSM700, Carl Zeiss, White Plains NY). Excitation lasers were 405 nm (DAPI), 488 nm (EGFP), and 561 nm (mCherry).

## Acknowledgements

The authors thank Zach Hall and India Nichols-Obande for review of the manuscript.

National Institutes of Health grant GM127260 (JCE). National Institutes of Health grant NS078410 (KNP). National Institutes of Health grant MD007602. National Institutes of Health grant HL103104 (supported BJB). National Institutes of Health grant HL007901 (supported EAA). National Institutes of Health grant HL117929 (supported CLG). National Institutes of Health grant HL116077 (supported AJB)

## Additional information

### Funding

| Funder | Grant reference number | Author |
| --- | --- | --- |
| National Institute of General Medical Sciences | GM127260 | J Christopher Ehlen |
| National Institute on Minority Health and Health Disparities | Pilot funding | J Christopher Ehlen |
| National Institute of Neurological Disorders and Stroke | NS078410 | Ketema N Paul |
| National Heart, Lung, and Blood Institute | Graduate Student Fellowship | Brittany J Bush |
| National Heart, Lung, and Blood Institute | Graduate Student Fellowship | Eva-Jeneé A Andrews |
| National Heart, Lung, and Blood Institute | Postdoctoral Fellowship | Cloe L Gray |
| National Heart, Lung, and Blood Institute | Postdoctoral Fellowship | Allison J Brager |
| National Institute on Minority Health and Health Disparities | MD007602 | J Christopher Ehlen |
| National Heart, Lung, and Blood Institute | HL103104 | Brittany J Bush |
| National Heart, Lung, and Blood Institute | HL007901 | Eva-Jeneé A Andrews |
| National Heart, Lung, and Blood Institute | HL117929 | Cloe L Gray |
| National Heart, Lung, and Blood Institute | HL116077 | Allison J Brager |

The funders had no role in study design, data collection and interpretation, or the decision to submit the work for publication.

### Author contributions

Brittany J Bush, Conceptualization, Formal analysis, Investigation, Methodology, Writing - original draft, Writing - review and editing; Caroline Donnay, Eva-Jeneé A Andrews, Formal analysis, Investigation, Methodology; Darielle Lewis-Sanders, Hamadi CS Brewer, Investigation; Cloe L Gray, Investigation, Methodology; Zhimei Qiao, Investigation, Methodology, Writing - review and editing; Allison J Brager, Hadiya Johnson, Investigation, Writing - review and editing; Sahil Sood, Software,

Methodology; Talib Saafir, Formal analysis, Visualization, Methodology; Morris Benveniste, Software, Formal analysis, Methodology, Writing - review and editing; Ketema N Paul, Conceptualization, Funding acquisition, Writing - review and editing; J Christopher Ehlen, Conceptualization, Formal analysis, Funding acquisition, Investigation, Methodology, Writing - original draft, Project administration, Writing - review and editing

## Author ORCIDs
Brittany J Bush ⓘ http://orcid.org/0000-0001-5168-5474
Eva-Jeneé A Andrews ⓘ http://orcid.org/0000-0003-1006-2152
Hadiya Johnson ⓘ http://orcid.org/0000-0002-8527-8313
Morris Benveniste ⓘ http://orcid.org/0000-0001-7070-1521
Ketema N Paul ⓘ http://orcid.org/0000-0003-0226-9559
J Christopher Ehlen ⓘ http://orcid.org/0000-0003-3223-9262

## Ethics
This study was performed in strict accordance with the recommendations in the Guide for the Care and Use of Laboratory Animals of the National Institutes of Health. All of the animals were handled according to a protocol (21-02) approved by the Morehouse School of Medicine institutional animal care and use committee (IACUC). All surgery was performed under isoflurane anesthesia, and analgesia was provided. Every effort was made to minimize pain and suffering.

## Decision letter and Author response
Decision letter https://doi.org/10.7554/eLife.80206.sa1
Author response https://doi.org/10.7554/eLife.80206.sa2

## Additional files

### Supplementary files
• MDAR checklist

### Data availability
Data generated in this study are deposited in Dryad.

The following dataset was generated:

| Author(s) | Year | Dataset title | Dataset URL | Database and Identifier |
|---|---|---|---|---|
| Ehlen JC | 2022 | Non-rapid eye movement sleep determines resilience to social stress | https://dx.doi.org/10.5061/dryad.x0k6djhn4 | Dryad Digital Repository, 10.5061/dryad.x0k6djhn4 |

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
