## [Editor Report]

This well-written, convincing report provides new insights for neuroscientists studying sleep architecture and stress sensitivity. A particularly important conclusion is that differences in sleep architecture before chronic social defeat stress may serve as a predictive biomarker of stress resilience.

---

## [Decision Letter]

**Decision letter after peer review:**

Thank you for submitting your article "Non-rapid eye movement sleep determines resilience to social stress" for consideration by *eLife*. Your article has been reviewed by 3 peer reviewers, and the evaluation has been overseen by a Reviewing Editor and Kate Wassum as the Senior Editor. The following individual involved in the review of your submission has agreed to reveal their identity: Ana Pocivavsek (Reviewer #3).

Essential revisions:

The authors all agree that the manuscript has sufficient data in it for publication, so no additional experiments are required. There was agreement across all reviewers, however, that the authors needed to consider several points and that the manuscript required some significant changes and additions to contextualize and clarify their data. These comments are all listed below:

1. A major question is where and when these individual differences in stress are encoded in the brain. Unfortunately, determining if these differences exist before exposure to stress is not possible with the methods employed here, unless the case can be made that the surgical procedures required to obtain the sleep measurements are not stressful. This issue must be acknowledged in the report and more care needs to be given to ensuring that the terminologies used are always clear and accurate (i.e., "before stress" should be "before social defeat stress").

2. In Figure 2: Increasing sleep by activating the preoptic area is a clever and novel approach that deserves more attention and promotion in this manuscript. Describing this method further, whether in this manuscript or another, can include electrophysiological validations of how the Gq DREADD is affecting preoptic neuronal firing and activity and whether a Gi DREADD has the opposite effect. The present version, however, is not optimally controlled. CNO is used instead of the more selective ligands compound 21 or deschloroclozapine (DCZ). Moreover, it is administered systemically rather than through a cannula directly to the preoptic area. Does CNO itself have any effect on the proportion of resilient vs. susceptible animals after CSDS? Even without considering those variables, an improved version of this experiment and figure would include a CNO-only condition and a DREADD+CNO+ non-defeated, control group. In light of this, the results describing this figure may overreach by claiming simply that "sleep is both necessary and sufficient for resilience to social-defeat stress." All susceptible animals in other published studies also slept, and more specifically, activating the preoptic area promoted resilience in this study. It may be worth considering whether another method of promoting sleep or activating another brain region would have the same effect.

---

## [Author Response]

Essential revisions:1. A major question is where and when these individual differences in stress are encoded in the brain. Unfortunately, determining if these differences exist before exposure to stress is not possible with the methods employed here, unless the case can be made that the surgical procedures required to obtain the sleep measurements are not stressful. This issue must be acknowledged in the report and more care needs to be given to ensuring that the terminologies used are always clear and accurate (i.e., "before stress" should be "before social defeat stress").

We have changed the text to show that sleep recordings were obtained before social-defeat stress. We have also acknowledged this issue in the discussion at line 411.

2. In Figure 2: Increasing sleep by activating the preoptic area is a clever and novel approach that deserves more attention and promotion in this manuscript. Describing this method further, whether in this manuscript or another, can include electrophysiological validations of how the Gq DREADD is affecting preoptic neuronal firing and activity and whether a Gi DREADD has the opposite effect. The present version, however, is not optimally controlled. CNO is used instead of the more selective ligands compound 21 or deschloroclozapine (DCZ). Moreover, it is administered systemically rather than through a cannula directly to the preoptic area. Does CNO itself have any effect on the proportion of resilient vs. susceptible animals after CSDS? Even without considering those variables, an improved version of this experiment and figure would include a CNO-only condition and a DREADD+CNO+ non-defeated, control group.

At this time, we do not have ephys data to validate the change in POA firing with Gq-DREADD activation.

We agree that investigating the effects of a Gi DREADD could be informative, however, we have not conducted these studies.

When these studies began, we were only aware of a few reports using compound-21 or deschloroclozapine as a DREADD agonist, therefore, we decided to use the well-established agonist CNO. Although CNO was in widespread use, we were also concerned by reports of off-target effects. Because our question centered around the interaction of sleep and social-stress, we chose to keep social-defeat stress and all other conditions constant while changing only sleep/VLPO activation. Thus, all animals in the study received social-defeat stress, virus injections and CNO. We felt that this study design did provide an adequate control for the use of CNO. Notably, in the cohort receiving the control viral construct, we observed an equal number of susceptible and resilient mice after CNO treatment. We agree that it would be ideal to have CNO-only and CNO+non-defeated controls and have added a discussion of the need to determine the effect of these treatments on social avoidance (line 471). In addition, we plan on continuing to refine this approach and to investigate more precise methods of delivery, other compounds, inhibitory DREADD, electrophysiological responses and additional methods of promoting sleep.